# Effect of Hydrogen on High Cycle Fatigue Properties of L360 Pipeline Steel Notched Specimens

**DOI:** 10.3390/ma17225612

**Published:** 2024-11-17

**Authors:** Liangliang Huang, Lin Zhang

**Affiliations:** College of Materials Science and Engineering, Zhejiang University of Technology, Hangzhou 310014, China; huang444973807@163.com

**Keywords:** hydrogen embrittlement, notched fatigue life, fatigue crack extension, stress intensity factor ΔK

## Abstract

The fatigue characteristics of notched specimens of L360 pipeline steel in hydrogen and nitrogen environments were investigated by high cycle fatigue life tests and fatigue crack growth rate tests. The fracture morphology in the nitrogen environment was dominated by microcracks and fatigue strips. The fatigue fracture had distinctly different regions in the hydrogen environment. The outer region of the fracture in the hydrogen environment was similar to the nitrogen environment, but a large number of hydrogen embrittlement features were found in the inner region. The fatigue crack growth rate tests were analyzed in conjunction with fatigue life tests. It was found that more fatigue cycles were required to achieve the stress intensity factor ΔK for rapid hydrogen-promoted crack propagation at lower stress. The region with hydrogen embrittlement features increases with decreasing stress.

## 1. Introduction

Due to worldwide dependence on fossil fuels, problems such as environmental pollution and the gradual depletion of oil resources have arisen [1,2,3]. Protecting the environment and reducing reliance on fossil fuels are becoming major global concerns. As a renewable, efficient, and clean energy source with many advantages [4,5,6], such as abundant resources and a wide range of sources, hydrogen energy is gradually gaining attention. Internationally, the United States, Europe, and other countries have incorporated hydrogen energy into their national energy development strategies. Transporting hydrogen by pipeline is the safest and most economical way [7,8]. However, transporting hydrogen will form an alternating load on the pipeline steel due to the pressure fluctuation, which can seriously affect pipeline steel fatigue life. Therefore, investigating the fatigue life of pipelines is crucial in the hydrogen environment.

Hydrogen is known to increase the hydrogen embrittlement susceptibility of pipeline steels [9,10,11,12,13,14]. Among these factors, hydrogen concentration, magnitude of load, and the stress concentration factor of pipeline steels affect the hydrogen embrittlement of the material. Gaddam et al. [15] investigated the effect of hydrogen on the expansion of fatigue cracks from a microstructural point of view. They showed that the fatigue life of the material was significantly decreased in the hydrogen environment. Briottet et al. [16] studied the effect of crack sprouting and crack initiation on specimens under different hydrogen pressure conditions and finally analyzed the fatigue life.

Notches in pipeline steels can form stress concentrations that cause local force changes and affect material fatigue life [17,18,19]. Under such conditions, the material will preferentially form stress concentrations at the notches, which will be exacerbated by hydrogen, affecting the material’s mechanical properties in the hydrogen-containing environment. In previous studies on hydrogen embrittlement of notched specimens, Meng et al. [20] studied the hydrogen embrittlement sensitivity of X80 pipeline steel under different hydrogen/natural gas mixture conditions. Zhou et al. [21] studied the tensile properties and morphology of X80 pipeline steel with and without notched conditions. They found that hydrogen had a significantly increased effect on the mechanical properties of the specimens. Nagaishi et al. [22] investigated the fatigue life of notched stainless steel 304 under air and hydrogen environments and analyzed the differences between the specimens in different environments by finite element analysis. An et al. [23] studied the property of notched specimens’ fatigue under various hydrogen pressures. They found that fatigue crack sprouting was more important in reducing fatigue life as the hydrogen partial pressure increased. Therefore, it is essential to investigate the fatigue life of notched specimens in the hydrogen environment. In the presence of a notch in the specimen, stress concentrations are formed at the notch. When hydrogen is present, it promotes crack propagation at the notch and ultimately affects the fatigue life of the specimens. In previous studies, there was no mention of how hydrogen affects fatigue cracking at the specimen notch during high-frequency fatigue cycles. Since minor notches can exist during the actual hydrogen transportation process, it is crucial to study the fatigue life of notched specimens in hydrogen environments.

This study investigated the high cycle fatigue properties of notched specimens of L360 pipeline steel in hydrogen and nitrogen environments. Through scanning the fracture morphology and analyzing the fatigue behavior of the specimen fracture, combined with the analysis of the relationship between the fatigue crack growth rate and the stress factor ΔK, it was found that hydrogen significantly promotes stage II of the fatigue crack growth rate, and the hydrogen embrittlement feature is concentrated in this region. As the stress decreases, more fatigue cycles are required to achieve the stress intensity factor ΔK for rapid hydrogen-promoted crack expansion, and the region of crack initiation and expansion increases as the stress decreases.

## 2. Materials and Methods

### 2.1. Experimental Materials

This study on the chemical composition testing of L360 pipeline steel was carried out by X-ray fluorescence spectrometry according to the standard GB/T 223.79 [24]. The chemical composition of L360 pipeline steel tested by X-ray diffraction is shown in Table 1. The dimensions of the test specimens are shown in Figure 1. Experiments were conducted using the INSTRON 8801 servo-hydraulic press (Norwood, MA, USA) at room temperature in a sealed environment under nitrogen and hydrogen environments. The gas pressure was 6 MPa for all tests. Experiments were carried out using 100#, 200#, 400#, 800#, 1200#, and 2000# sandpaper for surface preparation, followed by full polishing. Figure 2 shows a schematic diagram of the fatigue test equipment and test system. In this study, the data from the tensile test, fatigue life test, and fatigue crack extension test were analyzed descriptively to show the distribution and characteristics of the data by plotting linear plots and histograms. The effect of hydrogen on fatigue life was also analyzed by scanning the fracture morphology.

### 2.2. Slow Strain Rate Tensile Test

Tensile test with reference to the standard GB/T 228.1-2021 [25] for testing. Uniaxial stretching of round the bars specimens was performed by displacement control in nitrogen and hydrogen environments. The surfaces of all specimens were sufficiently polished to eliminate the effects caused by machining marks and then ultrasonically cleaned in acetone. The test rate was 2.0 × 10^−5^ s^−1^. The yield and tensile strengths from the tensile tests were used as the basis for parameter selection for the subsequent fatigue life tests. Each group of trials was conducted 3 times.

### 2.3. Fatigue Life Test

High-frequency fatigue life test with reference to the standard GB/T 6398-2017 [26] for testing. The effect of hydrogen on the high cycle life of L360 notched specimens under different applied stresses was investigated. The radius of curvature of the notched specimens at the root of the notch was 0.25 mm. The maximum applied stress loads were 90%, 80%, and 70% of the tensile yield strength (σmax/σy = 0.9, 0.8, 0.7). Each group of trials was conducted 3 times.

### 2.4. Fatigue Crack Growth Rate Test

Fatigue crack growth rate (FCGR) tests were ground sufficiently before testing to eliminate the influence of machining marks on the FCGR tests. Fatigue crack growth rate test with reference to the standard GB/T 6398-2017 for testing. Before the test, the specimens were pre-cracked, and the pre-cracking process was carried out by the reducing ΔK method. During the pre-cracking process, the stress ratio was 0.1 and the frequency was 10 Hz. The pre-cracking process was ended when the cracks extended to 6 mm and ΔK was 7 MPa·m^1/2^ at this crack length. The fatigue crack growth rate tests were performed at the end of pre-cracking and the stress ratio was 0.1. The amplitude of the force used in the experimental procedure was 1.5 kN and the frequency was 1 Hz. The experiment was stopped when the crack extended to 16 mm. The relationship curves between fatigue crack growth rate and stress intensity factor ΔK were obtained. Each group of trials was conducted 3 times. Zafra [27] found that the parent material exhibited martensitic slat debonding while the heat affected zone experienced martensitic slat debonding and intergranular fracture by examining fatigue crack extension tests in pre-hydrogenated and air environments.

## 3. Results

### 3.1. Slow Strain Rate Tensile Test

The stress–strain curves of the L360 pipeline steel tensile specimens are shown in Figure 3. The specimens showed lower elongation after rupture and reduced area in the hydrogen environment.

In the nitrogen environment, the yield strength was 498 MPa, the ultimate tensile strength was 593 MPa, and the elongation after rupture was 27.3%. In the nitrogen environment, the yield strength was 506 MPa, the ultimate tensile strength was 608 MPa, and the elongation after rupture was 29.8%. To subsequently facilitate the setting of test parameters, the maximum loading stress loads of 90%, 80%, and 70% of the yield strength were selected as 450 MPa, 400 MPa, and 350 MPa, respectively.

### 3.2. Notched Fatigue Life Test

The S-N curves for the notched fatigue life test performed are shown in Figure 4. With decreasing stress loads, the fatigue life of notched specimens in the hydrogen environment was about 64%, 76%, and 83% of that in the nitrogen environment. The effect of hydrogen on the fatigue life decreases with the decrease in stress.

### 3.3. Fatigue Crack Growth Test

The da/dN-ΔK curves under constant load conditions are shown in Figure 5. Fatigue crack growth rates are similar in hydrogen and nitrogen environments when ΔK is lower than 14 MPa·m^1/2^. When ΔK reaches 14 MPa·m^1/2^, the fatigue crack growth rate significantly increases in the hydrogen environment. At ΔK above 21 MPa·m^1/2^, the cracks extended steadily inside the specimens. Compared with the specimens in the nitrogen environment, hydrogen mainly promoted the fatigue crack growth rate in stage II.

### 3.4. Fracture Morphology

Figure 6 shows the fracture morphology of notched fatigue specimens under different stress conditions in hydrogen and nitrogen environments. Figure 6a–c show the fracture morphology under different stress loading conditions in the nitrogen environment; Figure 6d–f show the fracture morphology under different stress loading conditions in the hydrogen environment. It can be observed that there are two distinctly different regions of the L360 fracture specimens in the hydrogen environment, which are divided into Region I and Region II. It can be noticed that Region I increases gradually with the decrease in stress. The fracture morphology under the highest and lowest applied stress loads is subsequently analyzed.

Figure 7 shows the fracture morphology of notched specimens stressed in the nitrogen environment. There are a large number of fatigue stripes and microcracks in the specimens in the nitrogen environment. As the fatigue cycle progresses, the number of fatigue stripes in the direction of fatigue crack extension in the specimens increases, as shown in Figure 7c,f. Moreover, as the stress decreases, there are fewer microcracks and more fatigue stripes in the fracture, as shown in Figure 7e,f.

Figure 8 shows the fracture morphology of the fatigue life test specimens in the hydrogen environment. Compared to the specimens in the nitrogen environment, the fracture in the hydrogen environment shows two distinctly different regions. To compare the differences between the two regions, a distance of 0.5 mm around the division of the two regions was taken for characterization and analysis. The fracture was found to have little difference in morphology between the hydrogen and nitrogen environments in Region I, which was mainly characterized by microcracks and fatigue stripes, as shown in Figure 8a,c. However, in Region II, it was found to be characterized by hydrogen embrittlement, which was mainly characterized by the presence of a large number of quasi-cleavage surfaces, as shown in Figure 8b,e.

When the stress is reduced to 350 MPa, Figure 8f shows fewer quasi-cleavage regions than Figure 8c. This is consistent with the transition from Stage I to Stage II in the fatigue crack growth test, where there is a turning point for hydrogen-accelerated crack extension. The large amount of quasi-cleavage surfaces in Region II indicates that hydrogen mainly influences stage II of the fatigue crack growth rate, which significantly enhances the rapid initiation of fatigue cracks. Compared to 450 MPa stress, there are fewer quasi-cleavage regions under 350 MPa stress conditions, indicating that the effect of hydrogen on fatigue life decreases with decreasing stress.

## 4. Discussion

The characterization of fracture morphology can be shown in that there are distinctly different regions in the hydrogen environment, as shown in Figure 6d–f. The obvious transformation of the morphology in Region I and Region II is consistent with the change of fatigue crack growth rate in the hydrogen environment. The curves of Figure 9, displacement amplitude versus the number of cycles, illustrates that during the fatigue life cycles there exists a stage of rapid crack expansion, which is accelerated by the presence of hydrogen. The point of rapid crack initiation is determined by the intersection of two lines, one line is almost horizontal due to fatigue crack initiation and expansion, and the other line has a large change in slope due to rapid expansion of fatigue cracks, which is illustrated in Figure 9a–c. Birenis et al. [28] studied the fatigue crack growth behavior at the crack tip of pure iron and concluded that the beginning of stage II was the quasi-cleavage fracture. There is a close correlation between the formation of quasi-cleavage surface features and the acceleration of the fatigue crack growth rate in hydrogen, which indicates that hydrogen has a significant enhancement effect on the fatigue crack growth rate, which is one of the reasons for the reduction of the fatigue life in the hydrogen environment.

Fatigue life experiments are divided into three stages: fatigue crack initiation, extension, and final fracture. Compared to unnotched specimens, notched specimens will have multiple sources of fatigue due to stress concentrations [29]. Small cracks sprout on the surface of the notched root, and these small cracks may not be on the same surface. In response to the application of stress, small cracks repeatedly open and close, forming a step that expands in the direction of the final fracture. The fatigue stripes observed at 450 MPa were characterized in hydrogen and nitrogen environments. It was found that the fatigue stripes reveal different directional orientations in various regions, ultimately towards the direction of tearing. The above phenomenon can be observed in Figure 10a.

When the specimens were tested in the hydrogen environment fatigue stripes and microcracks were mainly observed in Region I, and the direction of fatigue stripes in Region I under hydrogen matched the direction in the nitrogen environment. This indicates that the notched fatigue specimens cracked in all directions from the surface during the test, and that there were multiple crack sources, as shown in Figure 10(b-1–b-5). To inhibit crack extension, microcracks appear inside the specimens to consume the energy generated by the cyclic load applied to the specimens. It can be observed in Figure 10 that there are significantly more microcracks in Region I in the hydrogen environment than in the nitrogen environment. As the stress decreases, the energy generated inside the specimens also decreases, and fewer microcracks are generated, so Region I increases with decreasing stress in the hydrogen environment in Figure 7b,c. In Region II, the hydrogen embrittlement feature is mainly present in this region due to the promotion of crack extension by hydrogen, as shown in Figure 10(b-6).

The results of the fatigue crack growth rate relationship between the fatigue crack growth rate and the stress intensity factor ΔK range for L360 pipeline steel CT specimens in hydrogen and nitrogen environments are shown in Figure 6, which reveals that fatigue crack growth rate in the hydrogen environment has an obvious rapid increase, which is the stage II in Figure 6. This is consistent with the relationship between displacement amplitude and fatigue life in the fatigue life test. The fatigue life consumption at each stage under three different stress amplitude conditions in the hydrogen and nitrogen environments can be observed from Figure 11. In stage I, the number of cycles for fatigue crack initiation and extension increases significantly with decreasing stress load. The presence of hydrogen promotes rapid fatigue crack extension and this effect decreases with stress reduction. Hydrogen promotes rapid crack extension but is required to achieve the desired stress intensity factor. When the stress decreases, the initial crack growth rate is relatively slow and needs to be formed by repeated crack opening and closing. So as the stress decreases, the specimens need to be subjected to more fatigue cycles to achieve the stress intensity factor required for hydrogen to promote rapid crack extension in the hydrogen environment. This explains why the region of Stage I increases with decreasing stress.

Neeraj et al. [30] proposed a mechanism based on hydrogen accumulation leading to local plastic deformation, hydrogen vacancy damage, and induced hydrogen embrittlement by observing ferritic steels’ hydrogen embrittlement fracture characteristics, as shown in Figure 12. The present study is also based on this mechanism, proposing that the hydrogen-assisted fatigue process ultimately affects fatigue life through these four stages. Since the specimens are notched, there are stress concentrations at the notch. When loading in Stage I, the material cracks at the notch and generates plastic dislocations (Stage II). However, in the presence of hydrogen, hydrogen will accumulate and build up from the environment towards the high stress at the notch, forming a region that favors plastic deformation, the region defined by Neeraj et al. [30] as the HELP region. In stage III, as the fatigue test proceeds, superfluous vacancies will be formed, which accumulate in large numbers in the HELP region, and these vacancies will combine with hydrogen, eventually forming nucleation and growth of nano-gaps in stage IV, leading to crack generation. An et al. [31] investigated the ease of crack formation and extension at the notch under different horizontal stress concentration conditions. In this study, the effect of stress magnitude on the HELP region at the notch was investigated in the direction of stress conditions and at the same stress level, and it was found that the effect of hydrogen on this region was different. Since most of the fatigue fracture process is concentrated in the vacancies formed in the third stage, two distinct regions are found at the fracture. The transition from the third to the fourth stage is the stage of rapid growth of the displacement amplitude Da value in Figure 11. However, the number of fatigue cycles with rapidly increasing values of displacement amplitude Da in the hydrogen environment approaches the specimens in the nitrogen environment with decreasing stress, which indicates that the effect of hydrogen on the rapid expansion of cracks at notches decreases with decreasing stresses. Zeng et al. [32] investigated the effect of a hydrogen environment on the fatigue fracture morphology of X80 pipeline steel. Combined with SEM analysis of the fracture morphology, the decrease in the size and density of the tough nests decreased the displacement amplitude, while the increase in the plane area increased the displacement at fatigue fracture due to the accelerated crack extension. M.A. Mohtadi-Bonab [33] investigated the effect of different parameters on hydrogen-induced fatigue failure of pipeline steels and found that the effect of hydrogen on the rate of fatigue crack extension is more pronounced because hydrogen atoms migrate to the crack tip at low frequencies. Fatigue crack extension accelerates with increasing hydrogen pressure, and its extension is minimum when tested in air. Li et al. [34] established a CPFEM model based on the principles of crystal plasticity theory and proposed a fatigue life prediction model for pre-charged hydrogen austenitic stainless steel. The model successfully predicted the fatigue life under different hydrogen contents and stress amplitudes, and the predicted fatigue life was well fitted to the S-N curve within two times error.

## 5. Conclusions

In this study, the high cycle fatigue performance of notched specimens of L360 pipeline steel was investigated in hydrogen and nitrogen environments. Based on experimental data and fracture surface morphology analysis, the following conclusions were drawn:As the stress decreases, the fatigue life under the hydrogen environment gradually approaches that of the specimens under the nitrogen environment, which indicates that the effect of hydrogen on the fatigue life of notched specimens decreases with the increase in stress.The fatigue life specimens’ fracture has two significantly different areas in the hydrogen environment, and the area of Region Ⅰ increases as the load stress decreases. This is mainly due to the fact that with the decrease in stress, the specimens need to be subjected to more cycles to achieve the stress intensity factor ΔK required for hydrogen to promote rapid crack extension.The fracture in the hydrogen environment has little difference in Region Ⅰ from the specimens in the nitrogen environment. It indicates that the presence of hydrogen does not have much effect in the process of crack initiation and gradually expanding in stage I. A large number of hydrogen embrittlement features were found in Region II, which suggests that hydrogen mainly increases the fatigue crack extension rate in stage II and significantly reduces the total fatigue life in the hydrogen environment.

## Figures and Tables

**Figure 1 materials-17-05612-f001:**
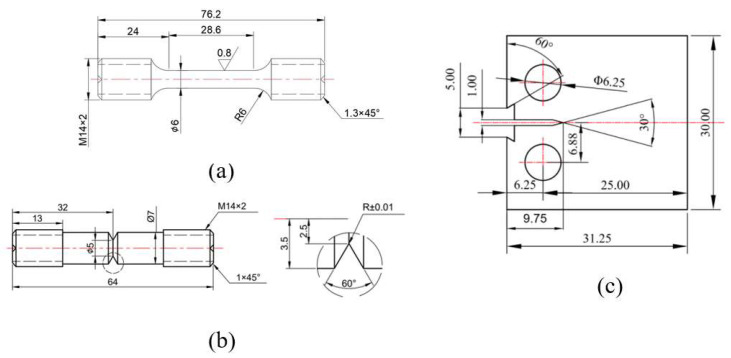
Dimensions of the test specimens: (**a**) tensile test; (**b**) notched fatigue life test; (**c**) fatigue crack growth rate test.

**Figure 2 materials-17-05612-f002:**
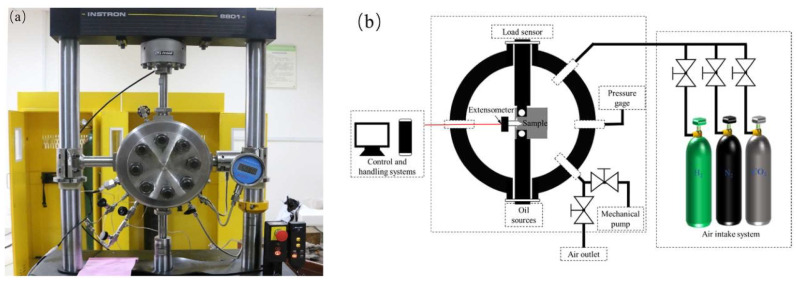
Schematic diagram of fatigue test equipment. (**a**) Instron fatigue testing machine; (**b**) experimental system schematic diagram.

**Figure 3 materials-17-05612-f003:**
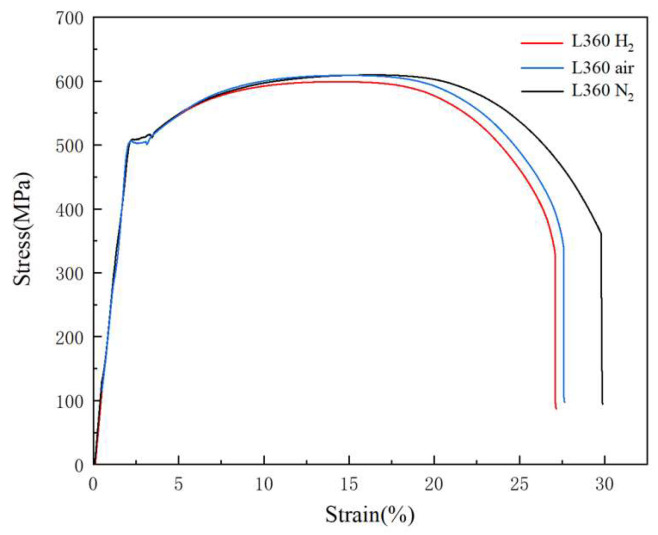
Stress–strain curves of L360 pipeline steel tensile specimens in hydrogen and nitrogen environments.

**Figure 4 materials-17-05612-f004:**
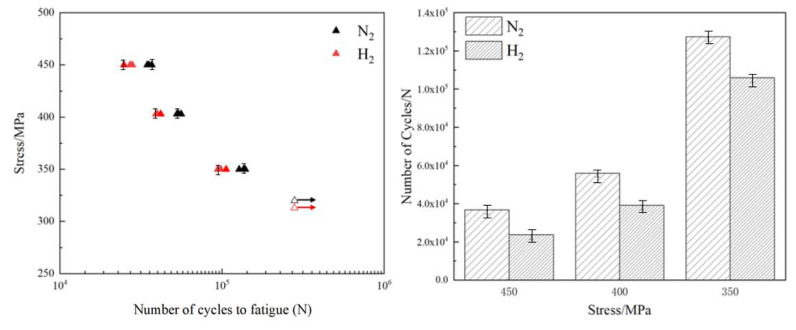
Fatigue life of notched specimens of L360 pipeline steel in hydrogen and nitrogen environments.

**Figure 5 materials-17-05612-f005:**
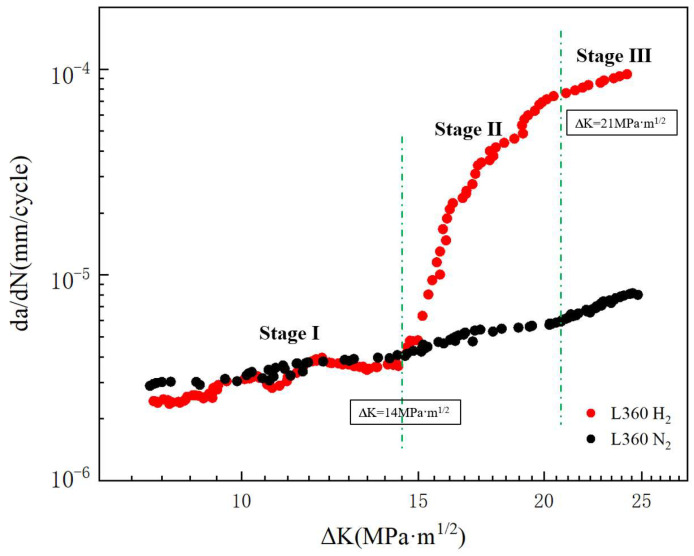
FCGR curves of L360 pipeline steel in 6 MPa hydrogen and nitrogen environments.

**Figure 6 materials-17-05612-f006:**
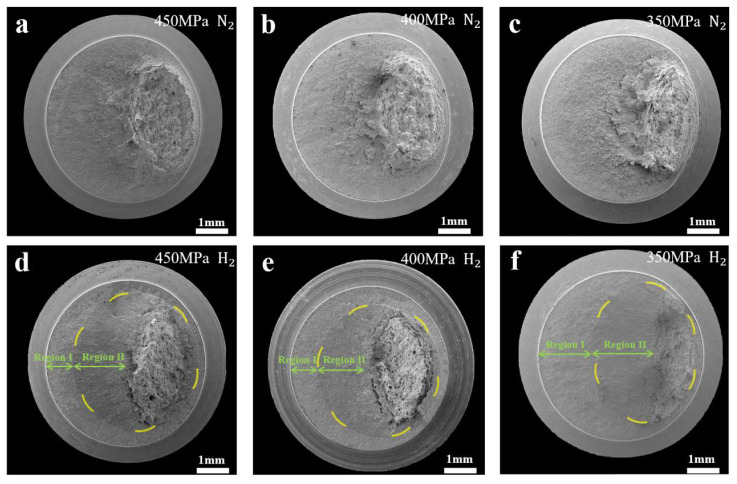
Fracture morphology of notched fatigue specimens under different stress loads: (**a**–**c**) nitrogen gas; (**d**–**f**) hydrogen gas.

**Figure 7 materials-17-05612-f007:**
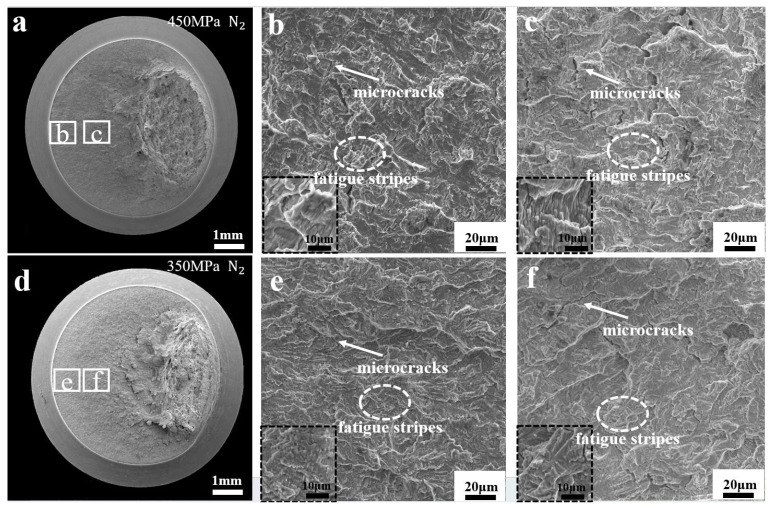
Fracture morphology of notched fatigue specimens with stresses of 450 MPa and 350 MPa in the nitrogen environment: (**a**–**c**): fracture morphology under 450 MPa stress in nitrogen environment, (**b**) is the fracture morphology under low magnification, (**c**) is the fracture morphology under high magnification; (**d**–**f**): fracture morphology under 360 MPa stress in nitrogen environment, (**e**) is the fracture morphology under low magnification, (**f**) is the fracture morphology under high magnification.

**Figure 8 materials-17-05612-f008:**
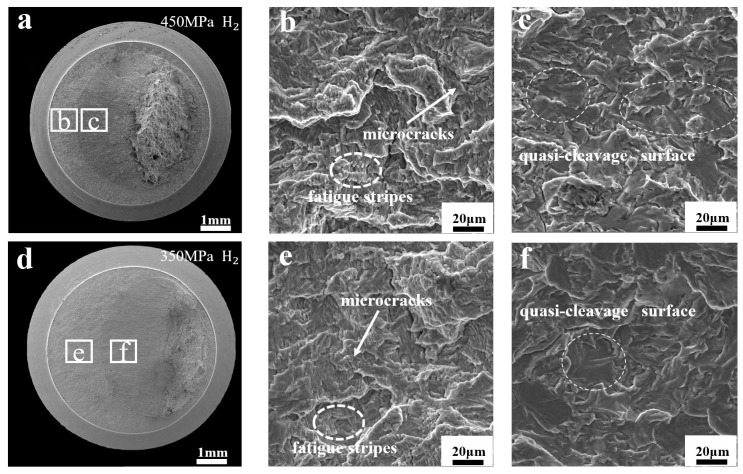
Fracture morphology of notched fatigue specimens with stresses of 450 MPa and 350 MPa in the hydrogen environment: (**a**–**c**): fracture morphology under 450 MPa stress in hydrogen environment, (**b**) is the fracture morphology under low magnification, (**c**) is the fracture morphology under high magnification; (**d**–**f**): fracture morphology under 360 MPa stress in hydrogen environment, (**e**) is the fracture morphology under low magnification, (**f**) is the fracture morphology under high magnification.

**Figure 9 materials-17-05612-f009:**
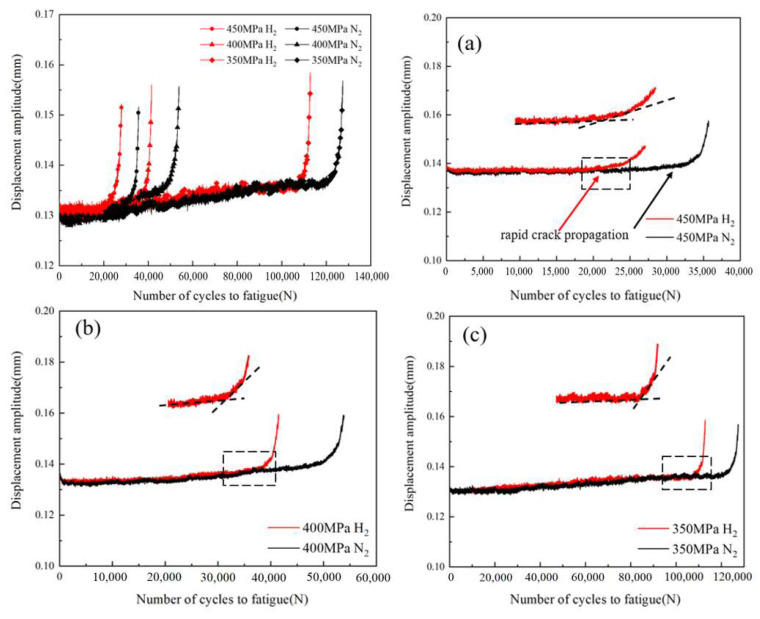
Curves of different stress load displacement amplitudes versus the number of cycles: (**a**) 450 MPa; (**b**) 400 MPa; (**c**) 350 MPa.

**Figure 10 materials-17-05612-f010:**
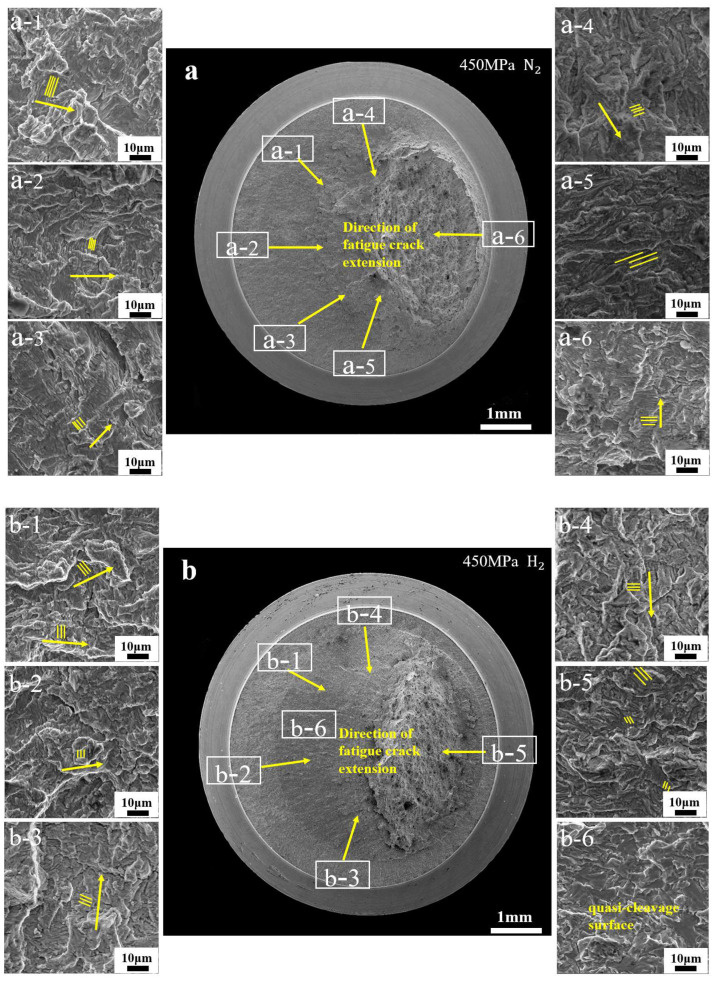
Fatigue life test under 450 MPa stress: (**a**) in a nitrogen environment; and (**b**) in a hydrogen environment. (a-1)–(a-6) are the enlarged fracture diagrams after fatigue life test under nitrogen; (b-1)–(b-6) are the enlarged fracture diagrams after fatigue life test under hydrogen.

**Figure 11 materials-17-05612-f011:**
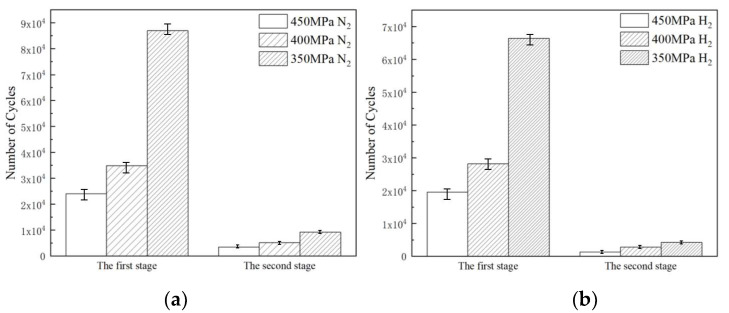
Fatigue life consumed at various stages under different stress amplitude conditions in hydrogen and nitrogen environments: (**a**) nitrogen environment, (**b**) hydrogen environment.

**Figure 12 materials-17-05612-f012:**
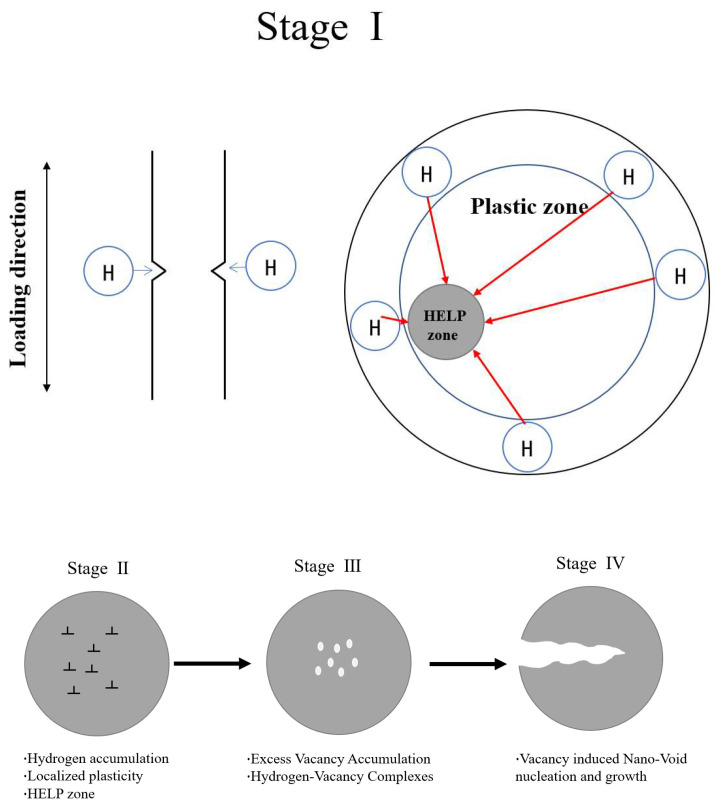
Schematic diagram of the mechanism of local plastic deformation, hydrogen vacancy damage, and induced hydrogen embrittlement based on hydrogen accumulation.

**Table 1 materials-17-05612-t001:** Chemical composition of L360 steel used in this study (wt%).

C	Si	Mn	P	S	V	Ti	Fe
0.24	0.45	1.40	0.025	0.015	0.1	0.04	balance

## Data Availability

The original contributions presented in the study are included in the article, further inquiries can be directed to the corresponding author.

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
