# Peer review of "Effect of Hydrogen on High Cycle Fatigue Properties of L360 Pipeline Steel Notched Specimens"

_materials, 2024, doi:10.3390/ma17225612_

Round 1

Reviewer 1 Report

Comments and Suggestions for Authors

This work presents high cycle fatigue properties of notched specimens of L360 pipeline steel in hydrogen and nitrogen environments.
Here are some comments to improve the paper:
-It would also be appropriate to include the dependences on pressure and temperature changes in the nitrogen and hydrogen environment in the tests.
-Page 3, line 82: On what basis was the speed of the test chosen?
-What was the surface roughness of the ground samples?
-What instrument was used to examine morphology?
-It would be appropriate to present a comparison of results from other works.

Reviewer 2 Report

Comments and Suggestions for Authors

My comments/suggestions are given hereafter:

1.       Title: Please also mention that you also investigated the effect of Nitrogen.

2.       Introduction: You need to discuss more in detail how this work complements existing literature. Also, what is the reason for studying the combination of hydrogen and nitrogen? This is important to strengthen the novelty of your work as you only mention that ‘it is essential to investigate the fatigue life of notched specimens in the hydrogen environment’.

3.       Materials and methods: Are the tensile and fatigue tests based on a standard? If so, please mention this.

4.       Materials and methods: A clear description of the surface preparation procedure should be given in the experimental part, as hydrogen diffusion is a surface phenomenon. Also, instead of mentioning ‘sufficient polishing’ you need to specify resulting roughness.

5.       Materials and methods: Repeatability of tests is not given.

6.       Results: Throughout the text, the spread of experimental values should be given along with the average value.

7.       Figure 2: To my point of view, the stress-strain curve for the as-received material (without any hydrogen and/or nitrogen uptake) should be included.

8.       Figure 3a: Please add error bars.

9.       Figure 3b: Error bars appear to be the same for all conditions (stress and environment). Is this correct?

10.   Figure 10: Error bars are missing.  

11.   Discussion: To my point of view, in the discussion there is no link between the material microstructure and hydrogen and/or nitrogen uptake. This is important as this journal focuses strongly on ‘Materials’.

12.   Discussion: Hydrogen failure mechanism should be discussed (e.g. formation of hydrides, HELP theory etc.).

Reviewer 3 Report

Comments and Suggestions for Authors

The manuscript entitled “Effect of hydrogen on high cycle fatigue properties of L360 pipeline steel notched specimens ” is in line with the Materials journal. The article is based on original research. The article has a proper composition but requires some changes and supplementation before publication, as follows:

·       Abstract: the abstract has to be rewritten: explain the motivation, clarify research methods and add measurable results.

·       Introduction: explain the research gap and stress the novelty.

·       Chapter 2.1. Add information about the manufacturer of the equipment used for the test (journal requirements).

·       Chapter 2.1. Specify the information about the chemical composition of the materials. What kind of research was applied to investigate it? Give the source of data or add information about the research methods used.

·       Chapter 2.1. How many samples have been tested?

·       Chapter 2.1. Add information about microstructural research – methodology is not described.

·       Chapter 2.1. What kind of statistical analysys was applied?

·       Discussion: The discussion should involve previous research from the last 5 years.

Round 2

Reviewer 2 Report

Comments and Suggestions for Authors

Dear authors,

After reading the updated version of the manuscript and your point-by-point reply to my comments, I now believe that this work is appropriate for publication.

Reviewer 3 Report

Comments and Suggestions for Authors

The manuscript entitled “Effect of hydrogen on high cycle fatigue properties of L360 pipeline steel notched specimens ” has been significantly corrected, However not all comments have been sufficiently adressed. The atricle requires further supplementation, as follows:

·       Chapter 2.1. Specify the information about the chemical composition of the materials - characterize the method (X-ray diffraction). Check if given method is correct.

·       Chapter 2.1. Add information about microstructural research – methodology is not described. It have to be presented in separate point.

·       Chapter 2.1. What kind of statistical analysys was applied? The information should be also included in the manuscript. Based on answer it is no proper statistical analysis in this article.

·       Discussion: The discussion should involve previous research from the last 5 years. The given literature is very old and it have to be supplemented about new positions.
